# Mechanistic Insight into Poly-Reactivity of Immune Antibodies upon Acid Denaturation or Arginine Mutation in Antigen-Binding Regions

**DOI:** 10.3390/antib12040064

**Published:** 2023-10-13

**Authors:** Tsutomu Arakawa, Teruo Akuta

**Affiliations:** 1Alliance Protein Laboratories, 13380 Pantera Road, San Diego, CA 92130, USA; 2Research and Development Division, Kyokuto Pharmaceutical Industrial Co., Ltd., 3333-26 Aza-Asayama, Kamitezuna, Takahagi-shi 318-0004, Ibaraki, Japan; t.akuta@kyokutoseiyaku.co.jp

**Keywords:** poly-reactivity, flexibility, domain, acid conformation, molten globule

## Abstract

The poly-reactivity of antibodies is defined as their binding to specific antigens as well as to related proteins and also to unrelated targets. Poly-reactivity can occur in individual molecules of natural serum antibodies, likely due to their conformation flexibility, and, for therapeutic antibodies, it plays a critical role in their clinical development. On the one hand, it can enhance their binding to target antigens and cognate receptors, but, on the other hand, it may lead to a loss of antibody function by binding to off-target proteins. Notably, poly-reactivity has been observed in antibodies subjected to treatments with dissociating, destabilizing or denaturing agents, in particular acidic pH, a common step in the therapeutic antibody production process involving the elution of Protein-A bound antibodies and viral clearance using low pH buffers. Additionally, poly-reactivity can emerge during the affinity maturation in the immune system, such as the germinal center. This review delves into the underlying potential causes of poly-reactivity, highlighting the importance of conformational flexibility, which can be further augmented by the acid denaturation of antibodies and the introduction of arginine mutations into the complementary regions of antibody-variable domains. The focus is placed on a particular antibody’s acid conformation, meticulously characterized through circular dichroism, differential scanning calorimetry, and sedimentation velocity analyses. By gaining a deeper understanding of these mechanisms, we aim to shed light on the complexities of antibody poly-reactivity and its implications for therapeutic applications.

## 1. Introduction

Antibodies are one of the most versatile reagents and are under extensive development for therapeutic uses [1,2,3]. The poly-reactivity of antibodies is one of the major concerns in their applications and is characterized by either enhanced antigen binding or non-specific binding to non-antigen proteins [4,5,6,7]. Poly-reactive antibodies can occur regardless of immunization, and this is likely due to the conformational flexibility of the antibodies and their ability to bind the target molecules, as well as off-targets, and to undergo domain–domain interactions [8,9]. The incorporation of arginine in the CDR has been found to augment the poly-reactivity [10,11,12]. In the absence of immunization by antigens, it has been observed that a portion of natural (naive) antibodies in serum is not active due to inhibition by binding to various specific or non-specific antigens [13,14]. In addition, a large number of monoclonal antibodies can bind to unrelated antigens, including endogenous host proteins [15]. A panel of naive monoclonal antibodies has shown binding to totally unrelated antigens, including different bacteria [16,17]. Poly-reactivity can be observed upon, for example, the acid-treatment of antibodies, which is encountered during antibody production comprising the elution of Protein-A bound antibodies and low pH viral clearance [7,18,19,20]. It can also be observed in protein denaturing or destabilizing conditions, including elevated temperature and treatments with redox, dissociating or denaturing agents [21,22]. Acid-induced poly-reactivity can be avoided through purification performed under neutral pH conditions, as summarized in our recent publication [23]. Poly-reactivity can also be augmented during the antibody maturation process, which may often involve the introduction of arginine mutations in the antigen-binding region (complementarity determining region, CDR) [24]. Thus, there may be three potential causes of poly-reactivity: the unmasking of the inactive antibodies, changes in the antibody conformation, and the increasing binding capacity through mutations. The last mechanism seems to be the least understood and, hence, we attempted to find a plausible explanation.

Antibodies have been shown to have high flexibility and dynamics [25] and undergo acid denaturation [19]. Acid denaturation is widely different from one protein to another [18,26]. For example, it has been observed with an IgG4 that this particular antibody undergoes gradual unfolding, retains a more or less tertiary structure, and has little tendency to aggregate in acid [18]. Other antibodies showed greater tendency to unfold and aggregate [27]. We show here that such acid denaturation results in enhanced flexibility in domain–domain interactions and in the different configuration of antibodies, which may be involved in poly-reactivity.

## 2. Poly-Reactivity

There are a few potential mechanisms, as described above, that cause antibody poly-reactivities. The first mechanism is the unmasking of inactive antibodies, as masking itself is an indication of poly-reactivity. It has been shown that such unmasking of the natural antibody population (i.e., in the absence of antigen immunization) can occur in vivo during a disease state or in vitro through heat treatment [13]. Antibodies undergo irreversible thermal denaturation with an onset melting temperature above ~55 °C depending on the heating rate [28]. Heating at mild temperature would be insufficient to cause the unfolding of each immunoglobulin domain, but may be sufficient to dissociate bound non-specific antigens (i.e., unmasking) or alter domain–domain interactions [14,29]. Urea was also used to induce the poly-reactivity of antibodies. For such a study, 6 M urea was used to treat human polyclonal IgG, resulting in a dramatic enhancement of the binding to its specific antigen [21]. Such a urea concentration could not only cause changes in domain–domain interactions, but also unfold each immunoglobulin domain [30]. Thus, the observed poly-reactivity induced by 6 M urea may be due to both reactions (an altered domain–domain interaction and domain unfolding) that are likely reversible, resulting in an altered antibody conformation.

Dissociating or chaotropic agents have also been used to induce poly-reactivity [21,22,31]. These conditions should be insufficient to denature proteins, but could dissociate bound inhibitors (unmasking) or alter domain–domain interactions. A high salt or glycine concentration has also been shown to cause the activation of antibodies in healthy individuals (without immunization) [14,31]. Such conditions are not known to cause the dissociation or denaturation of proteins. However, they could alter the electrostatic or polar interactions of antibody domains or antibody–antigen interactions. The unmasking of anti-cardiolipin antibodies in normal human serum was observed upon removing the antibody-bound phospholipids through phospholipase treatment [14].

Acid or alkaline treatments are common in causing poly-reactivity [14,21,31]. A number of reports have shown the increased binding of polyclonal IgG preparations to bacterial proteins and recombinant cytokines and monoclonal antibodies to their antigens after acid treatment, in particular when exposed to pH 2.8, as has been summarized [6]. The acid treatment of IgG also resulted in increased non-specific binding and enhanced suppression of sepsis by inhibiting inflammatory cytokines induced by the bacterial endotoxins (lipopolysaccharides), namely, cytokine storm [6]. They ascribed this enhanced binding to the exposure of hydrophobic regions by the acid treatment. They warned of the use of the acid exposure of antibodies during their preparation, as it might give false positives on antigen binding. Acid treatment has been shown to cause conformational changes in antibody structures that can lead to oligomer formation [18] and enhanced Fc receptor binding [19].

We have observed the non-specific binding of a low-pH-exposed antibody using Western blotting (unpublished observation), as has been reported [32]. Figure 1 shows such an example, the details of which are described in the figure legend. It compares a monoclonal antibody purified using a Protein-A/G column and conventional DEAE ion-exchange column. Western blotting was performed under exactly identical conditions, except for the antibody preparation [32]. Namely, the antibody was bound to a Protein-A/G column followed by elution at pH 2 or a conventional DEAE ion-exchange column run at a constant pH of 8.0. Figure 1 shows the Western blot of a cell lysate containing an antigen PLXDC2 (plexin domain containing 2) protein probed by its monoclonal antibody (developed by Abwiz Bio Inc., San Diego, CA, USA). The lane (-) corresponds to the SDS-PAGE of the control lysate without antigen expression. When the control lysate was probed by the Protein-A/G purified antibody (lane-2), many bands were detected by the antibody, most likely corresponding to non-antigen proteins. When the lysate with the antigen expressed (+, lane-3) was analyzed, a strong signal was obtained at the molecular weight of 75 k, corresponding to the PLXDC2 antigen. Thus, this antibody does bind specifically to the antigen protein. However, there are other bands stained by the antibody, indicating that the low-pH processed antibody preparation does bind non-specifically to foreign proteins, as seen with the control sample (lane-2). On the contrary, the DEAE-purified antibody showed much cleaner results, as shown in Figure 1. The control sample (lane-5) showed nearly no bands, indicating that the same antibody without low-pH exposure exhibited no non-specific binding to unrelated proteins. The antigen-containing lysate (lane-6) showed a much cleaner result with only a faint non-specific band above the strong staining of the antigen. It is interesting to point out that the staining of the antigen band appeared to be weaker with the DEAE-purified antibody (lane-6) than the Protein-A/G purified antibody (lane-3). These results indicate that the acid-processed antibody acquired the ability to bind to non-antigen proteins (non-specific binding) and also the ability to bind more strongly to the antigen protein. It should also be noted that the molecular weight markers were more strongly stained by the Protein-A/G purified antibody (compare lane-1 and lane-4), suggesting that this antibody bound to the marker proteins non-specifically, which added antibody-based staining to the already CBB-bound marker proteins.

Antibodies in serum have been shown to be inherently poly-reactive, likely due to their conformational flexibility [8,9]. It has been shown that the arginine content in CDR often increases in the antibody maturation process, which, in turn, augments the non-specific binding [33]. The arginine side chain has been shown to confer binding energies to inter- and intra-molecular interactions [34]. The contribution of four particular amino acids (Tyr, Ser, Gly and Arg) was examined for their effects on antibody affinity and specificity, showing that the Tyr makes the most important contribution to specific binding, while Gy and Ser makes the CDR region more flexible and Arg increases non-specific binding [35].

Poly-reactivity also occurs through the introduction of mutation to arginine residues in CDR as follows [24]. In this report, a correlation between non-specific binding and the CDR3 sequence of the heavy chains of multiple monoclonal antibodies raised against different antigens was examined. In general, a strong correlation was observed between the amount of arginine residues in the CDR3 and non-specific binding. The CDR3 of antibody variants against Aß peptide showed that those variants with more arginine residues showed higher non-specific bindings to the heterogeneous milk proteins and a penal of pure proteins [24]. Multiple clinical stage antibodies also showed the same correlation. Those monoclonal antibodies with CDR3 sequences that contain more basic amino acids (in particular, arginine) also showed higher tendency to self-associate, suggesting a correlation between non-specific binding and aggregation tendency [24].

Similarly, the number of arginine residues has been related to the non-specific binding of an antibody mimetic to Aß42 [11,12]. The antibody mimetic was generated by grafting potential self-recognition aggregation-prone sequences of Aß42, which included ^17^LVFFA^21^ or ^33^GLMVGGVVIA^42^ (Aß33-42), into the heavy-chain CDR3 of a model unrelated scFv construct [11,12]. With the latter construct containing self-associating Aß33-42 in the mimetic scFv, it was observed that a higher number of arginine residues, in particular when franked using hydrophilic amino acids, in the heavy-chain CDR3 increased the non-specific binding to the Aß42. However, it should be noted that this observation of the effects of arginine residues in the CDR3 sequence was made to enhance the self-association between the grafted Aß fragments and Aß42, which could be very different from the true complementary interaction between CDR and antigens.

## 3. Antibody Structure in Acid

Among the conditions used to identify poly-reactivities, the structure in acid is extensively characterized, while the effects on the antibody structure of mild temperature and chaotropic/dissociating agents are not well understood, but are unlikely to cause major conformational changes. Thus, we will focus on acid structures. Shown below is one of the detailed analyses of antibody conformation in low pH using model antibodies, i.e., two human monoclonal antibodies (hIgG4-A and -B) and a mouse monoclonal antibody (mIgG1), for which the antigen used to generate these antibodies was not disclosed in the original publication [18]. The near-UV CD spectrum of native hIgG4-A at pH 6.0 is characterized by a positive peak at 291~292 nm (attributable to tryptophan) and several negative peaks between 260 and 290 nm (261 and 268 nm attributable to tyrosine and phenylalanine), as seen in Figure 2A. The near-UV CD signals of the antibody at pH 2.7 or 3.5 are also shown in Figure 2A. These spectral features at low pH resembled those of the native protein at pH 6.0, as aromatic signals present in the native protein are also present in the spectra of low pH samples, as seen in CD signals at these characteristic wavelengths. It is also evident, however, that the CD intensities are shifted upward relative to the CD signals of the pH 6.0 sample (Figure 2A), although the 291 nm signal was unchanged at pH 3.5. A greater upward shift on these CD signals is observed at pH 2.7 than at pH 3.5, indicating a greater structure change. The presence of aromatic signals resembling those of the native protein indicates that, at pH 2.7 or 3.5, the protein retains a distinct tertiary structure and this further suggests that no gross conformational changes have occurred. Nevertheless, the observed upward shift of the CD signals relative to pH 6.0 does indicate conformational change at pH 3.5, which is further enhanced at pH 2.7. This upward shift might be attributed to changes in domain interactions, which could alter aromatic environments and, therefore, aromatic CD signals. These acid structures were highly stable [35].

The secondary structure of hIgG4-A at low pH was examined using the far-UV CD. The CD intensity at 217 nm, characteristic for the immunoglobulin-fold, is plotted in Figure 2A. No changes in this characteristic signal intensity were observed at pH 3.5, indicating no apparent change in the secondary structure at pH 3.5, despite the changes in the tertiary structure. This suggests that the observed change at pH 3.5 in the tertiary structure is not accompanied by a change in the secondary structure. This supports the notion that the observed changes in the tertiary structure are not due to the unfolding of ß-sheet domain fold, but due to the domain–domain interactions. The negative CD intensity at 217 nm is enhanced at pH 2.7, suggesting that the secondary structure is altered at this pH and the immunoglobulin-domain type β-sheet structure is increased. Thus, the observed greater near-UV CD change at pH 2.7 is accompanied by the change in the secondary structure. However, the observed slight increase in the secondary structure may be due to reduced domain–domain interactions, leading to the acquisition of a β-sheet structure in the flexible region.

The acid conformation was also studied for hIgG4-B at pH 2.9. As shown in Figure 2B, the hIgG4-B showed characteristic aromatic signals at 252, 258, and 291 nm at pH 2.9, similar to the native protein, although the intensities at pH 2.9 were shifted slightly upward relative to the pH 6.0 signals in a manner similar to hIgG4-A. In contrast, the CD features of the mIgG1 at pH 3.9 were almost identical to those at pH 6.0 (Figure 2C), indicating little conformational change at pH 3.9. Thus, it may be speculated that the response to acid treatment is highly variable and the conformational changes, if present, are largely restricted to domain–domain interactions.

Whether these acid structures are distinctly folded or unfolded at low temperatures can be evaluated using DSC. When folded, they should show cooperative melting in DSC analysis. As shown in Table 1, the native hIgG4-A showed a major endothermic peak at 78 °C (P_II_) and a minor peak (P_I_) at 67 °C. The observed biphasic thermal transition is typical for antibodies [36], reflecting the independent melting of variable and constant domains. At pH 3.5, the T_m_ of both peaks shifted to a lower temperature of 58 °C for P_II_ and 35 °C for P_I_. The decreased melting temperature can simply be attributed to general pH-induced destabilization. The thermal unfolding at pH 3.5 consists of two transitions, as in the native state, consistent with changes in domain packing, but not the unfolding of each domain at pH 3.5. At pH 2.7, the minor peak observed at pH 6.0 and 3.5 (P_I_) disappeared. This suggests the further destabilization of the protein structure and possibly further conformational changes, in particular for the domain responsible for the P_I_ transition, suggesting that a certain domain may have been unfolded. Thermal unfolding was irreversible at both pH 6.0 and 3.5, consistent with the turbidity observed with the samples recovered from the DSC cell, when heated over 100 °C.

Sedimentation analysis can also clarify the structure of antibodies. The sedimentation velocity is highly sensitive to changes in molecular size and shape. When the monomeric antibody aggregates, it sediments faster, and when it is unfolded, even in a single domain of multiple domain antibodies, it would sediment more slowly than the folded monomer due to the increased friction of the unfolded domain with the solvent through which the antibody sediments. Table 2 shows the results of the hIgG4-4. The sedimentation coefficient of the main peak, corresponding to the monomer, at pH 6.0 has a sedimentation coefficient, *s_20,w_*, of 6.74 S and represents 96.2% with some minor aggregate peaks present having sedimentation coefficients above 10.0 S. These aggregates are irreversible products, which do not dissociate upon dilution.

Table 2 also shows the size of the pH 3.5 sample. The main peak had a sedimentation coefficient of 7.08 S, a value slightly higher than the main peak at pH 6.0 (i.e., native monomer). This may be due to the different buffers used and, hence, different viscosities, but is not due to the aggregation at pH 3.5. Namely, the structure of this antibody at pH 3.5 is fully folded and is not unfolded even with its single domain. In addition to the main peak at 7.08 S (which is 97.5% of the total protein), there are minor peaks corresponding to aggregates above 10.9 S. There is also possibly a peak sedimenting more slowly than the monomer at ~3.5 S. Although the nature of this species is not entirely clear, it may represent an antibody fragment. One likely possibility is that this species is a half molecule, which could arise from the dissociation of antibodies at pH 3.5, in which the disulfide bond linking the two heavy chains has not been formed; such a half antibody could associate non-covalently to the native monomer. The amount of main peak (monomer) is essentially unchanged with time, even after 24 h incubation at pH 3.5, consistent with the observed stability of acid structures through the CD and DSC analysis.

Table 2 also shows the sedimentation coefficient at pH 2.7, which is not significantly different from that for the elution at pH 3.5, indicating that the structure of hIgG4-A is fully folded and is little changed at this pH. The main peak at 6.73 S is 97.7% of the total and, hence, is monomeric and stable. There appeared to be more half-antibodies.

CD clearly demonstrated some conformational changes at pH 3.5, although to a limited extent. The effects of the observed conformational changes on the pH-neutralized sample were examined using sedimentation analysis. When the pH 3.5 hIgG4-A was titrated with 1 M Tris-base to pH 6.0, the majority of the protein returned to the native state. Sedimentation analysis showed a monomer content of 97.5%, nearly identical to that of the original sample and the pH 3.5 sample (Table 2). There is little effect of incubation, at pH 3.5, on the monomer content of pH-titrated samples. The behavior of material titrated from pH 2.7 was quite different. As shown in Table 2, the monomer peak accounted for only 67%, with the balance as a broad range of soluble oligomers. These aggregates appeared to be stable. It may, thus, be concluded that weakened domain–domain interactions at pH 2.7 may have caused only the partial renaturation of the native domain–domain interactions and altered interactions may be responsible for oligomer formation.

Not all antibodies behave the same way in acid, as shown above. Surveying the literature, we find additional evidence of different acid structures. For example, a murine antibody CB4-1 and its fragment Fab showed extensive pH-dependent conformational changes through CD and fluorescence [37]. In more detail, the CB4-1, showed no conformational changes at pH 3.5 and new structures appeared between pH 3.5 and 2.0 [37]. At pH 3.1, the CB4-1 and its Fab fragment showed secondary and tertiary structures that are similar to a molten globule A-state, which is formed through a cooperative transition from the native structure. However, this A-state transition was irreversible, as its titration to neutral pH resulted in extensive aggregation [37] and, hence, it cannot be characterized by an equilibrium unfolding reaction. This appears to be different from the gradual multiple transitions observed for hIgG4-A. As an additional example, a mouse monoclonal antibody, IgG2b, raised against a human hemoglobin-β chain antibody, showed extensive acid-induced unfolding analyzed using CD and DSC, with only some secondary structure retained [28]. Namely, it gave only one endothermic peak of native IgG2b at pH 3.5 and no peaks at pH 2.0 in DSC analysis. It was speculated that the Fc fragment is first affected by low pH followed by the denaturation of the Fab fragment. With this mouse IgG2b, a significant β-sheet structure was lost even at pH 3.5, suggesting a possible unfolding of the domain structure, significantly different from the acid behavior of the IgG4-A and B as described above.

In another case, the MAK33 antibody showed a peculiar acid-induced conformation, designated as an alternatively folded state due to its unique thermal stability, which is different from the native state and which has a tendency to aggregate [38]. The above study demonstrated that, under acidic conditions (pH less than 3), the antibody MAK33 can assume a folded stable conformation, different from the native state and also different from the so-called A-state. The A-state is, in general, characterized by a high degree of secondary structure, increased hydrophobicity, and a lack of tertiary structure and, hence, no cooperative thermal transition [39,40]. On the contrary, this antibody MAK33 showed a native-like maximum wavelength of fluorescence emission, increased hydrophobicity, and a tendency toward slow aggregation. A prominent feature of this low-pH conformation of MAK33 is the stability against the denaturant and the cooperative thermal unfolding, indicative of the existence of well-defined tertiary contacts. The given data suggested that the MAK33 antibody at pH 2 adopted a cooperative structure that differs from the native immunoglobulin fold at pH 7, as well as from the A-state. This alternatively folded state exhibits certain characteristics of the molten globule A-state, but differs distinctly from the A-state in its extraordinary structural stability that is characteristic for native-like protein structures. These results suggest that this particular antibody assumes a folded domain structure even at pH 2.0, but with increased hydrophobicity and a tendency to aggregate, consistent with the loosening of domain packing, exposing hydrophobic surfaces that had been involved in domain–domain interactions. Certain properties, such as the presence of a tertiary structure, are similar to the above IgG4-A, but are different from the IgG4-A in its tendency to aggregate at low pH.

A completely different picture emerged with the CH3 domain of MAK33 [41]. The CH3 domains of antibodies are, in general, characterized by two antiparallel ß-sheets forming a disulfide-linked sandwich-like structure. At acidic pH values and low ionic strength, it was observed that the isolated CH3 of the MAK33 became completely unfolded, indicating that the domain–domain interaction of the MAK33 antibody stabilizes the folded structure of the individual immunoglobulin domain against low-pH-induced unfolding. The addition of salt caused a transformation of the acid-unfolded protein of the CH3 domain into an alternatively folded state, exhibiting a characteristic secondary structure, just as has been seen with the molten globule A-state [39,40,42,43,44,45]. Interestingly, this reaction involved the formation of a defined oligomer consisting of 12–14 subunits, clearly indicating that the surface properties of the salt-mediated folded structures of the A-state are different from those involved in native domain–domain interactions. This alternatively folded protein is remarkably stable against thermal and chemical denaturation and the unfolding transitions are highly cooperative [41].

A rabbit IgG showed four different conformations between pH 2 and 7 [19]. Below pH 5.5, it showed lowered domain–domain interactions, which resulted in independent tertiary structure loss of the CH2 domain below pH 3.0, which may be ascribed to the stabilization of this domain by domain–domain interactions. These acid-induced conformational changes resulted in the decreased affinity of the rabbit IgG for antigen binding and enhanced binding to Protein-A and C1q, a complementary component [19].

Mouse IgG2a (MN12) possessed a particular acid stability, showing non-random conformation even at pH 2.5 [46]. Its intrinsic fluorescence gradually increased with time at or below pH 3.42, above a pH of which no such fluorescence changes with time were observed. The observed increase reflected changes in the environments of fluorescent tryptophans. The more fluorescent MN12 antibody at low pH showed time-dependent aggregation, likely due to the exposed hydrophobic surfaces arising from the relaxed domain–domain interactions or domain unfolding. This change was irreversible, as the neutralization of the low-pH samples resulted in immediate precipitation due to their lower solubility at neutral pH. The absence of emission wavelength shift at low pH suggested that the fluorescent tryptophans have not been fully exposed to water, as seen in unfolding in guanidine hydrochloride, but rather that the partial quenching present in the native antibody has been removed by the weakened domain–domain interactions [46]. Namely, some fluorescent tryptophans are not fully accessible at neutral pH and become more accessible at lower pH, but are not accompanied by changes in tryptophan environments. The observed MN12 gradual conformational changes with time at lower pH are different from the low pH structures of hIgG4-A that retained the acid conformation for up to 10 days. Thus, it can be concluded that the way antibodies respond to low pH is highly variable, which may, in turn, lead to different conformations upon pH neutralization and, therefore, different specific antigen or receptor binding or different non-specific binding.

## 4. Antibody Flexibility

There are a number of reports demonstrating changes in domain–domain interactions in acid. Such changes in domain–domain interactions may reflect the flexible nature of the antibody structure. The development of technologies that can measure the flexibility and dynamics of antibodies at a single molecule level has been summarized [47,48]. Among them, cryo-electron (EM) tomography gives information on dynamic individual hydrated large structures, although it is limited to low resolution [49]. A mouse monoclonal IgG2a antibody was analyzed using cryo-EM tomography [50]. Low resolution structures were compared for four molecules of this antibody. The observed differences in conformation demonstrated flexibility in the hinge bending and rotational displacement of the Fab arms relative to the Fc. None of these four structures were identical in arrangements of the Fab and Fc region. The two Fab arms were not symmetrical relative to the Fc portion. The observed large conformational space implies that antibodies are inherently flexible.

Another technology is individual-particle electron tomography (IPET) [50]. Many individual structures with different orientations of the Fab and Fc domains showed the flexible and dynamic nature of the IgG1 antibody.

Hydrophobic interaction chromatography (HIC) suggested the possibility of its ability to detect the conformational ensemble of therapeutic monoclonal antibody IgG1 [51]. As depicted in Figure 3, the IgG1 was made 1 M in ammonium sulfate and bound to a HIC resin in 1 M ammonium sulfate. The bound IgG1 was eluted with descending ammonium sulfate gradient, resulting in the elution of five peaks. Among them, the first three peaks (IgG1-1, 2 and 3) were buffer-exchanged into a low salt buffer. Each of the three peaks was then made 1 M in ammonium sulfate and rechromatographed, leading to the appearance of three peaks (IgG-1, 2 and 3 in Figure 3) from each of the original peaks. This may be interpreted in terms of the conformational equilibrium of three IgG1 structures. As depicted in Figure 3, it was assumed that there are three different IgG1 structures (expressed as circle, square, and diamond) in the equilibrium. When the IgG1 was made 1 M in ammonium sulfate, its salting-out effect may freeze each of the three structures by reducing the rate of conversion between the three structures. Thus, each frozen structure is bound to the HIC column in the presence of 1 M ammonium sulfate and eluted depending on the hydrophobicity of each structure. Once eluted at low ammonium sulfate concentration, followed by buffer-exchange into the buffer, each structure is no longer stable and equilibrates with the other two structures. Rechromatography should then reproduce the above result. Although the properties of the above three structures are not clear, they may be due to the flexibility of domain–domain interactions, leading to a different degree of hydrophobic surface exposure. This difference is frozen by the strong salting-out effect of ammonium sulfate.

Another mechanism of poly-reactivity occurs during the antibody maturation process. There is unconventional strategy to diversify the repertoire of antibody specificities that potentially leads to poly-reactivity, which includes the conformational hetergoneity that goes far beyond those strategies adopted by somatic mutations [52]. One obvious cause of poly-reactivity is the flexible antigen binding regions due to large conformational spaces [52]. Another cause is the domain swapping of variable domains due to altered domain–domain interactions. Monoclonal antibodies against HIV-1 Env gp120 and gp41 showed broad antigen specificity upon mutations in the elbow region between the variable and constant domains [53]. The thermostability of the antibodies was associated with their specificity. The affinity gain for diversity in antigen recognition was coupled with decreased thermal stability. Key mutations in this maturation process occurred in the elbow region of their Fab fragments, which led to the increased flexibility of this region and concurrent reduced domain–domain interactions. These changes, in turn, resulted in a lower thermal stability and broader antigen specificity.

## 5. Relationship between Flexibility and Poly-Reactivity

An antibody comprises multiple domains with each domain folded into a typical ß-sheet structure. Figure 4 shows several potential antibody structures in acid due to protonation, having different domain packing in the Fab or Fc portion or both from the native structure and also having at least one domain losing the ß-sheet fold. These acid structures may lead to different structures upon pH neutralization and, therefore, different binding properties to the antigen and unrelated proteins, as has been observed. Arginine mutation also adds positive charges to the CDR regions and, hence, may have a similar effect on antibodies to the effects observed by protonation, as depicted in Figure 5. Namely, while arginine itself can cause different binding abilities to the antibody, it may affect the local CDR structure through intra- or inter-domain electrostatic and aromatic interaction, as depicted in Figure 5.

The bivalent mechanism of an antibody may play a role in its poly-reactivity. Figure 6 shows the normal bivalent binding of an antigen (panel A), in which two antigen molecules bind to both Fab arms with a free energy of ΔG_SP_. On the contrary, Figure 6B depicts the weaker binding of an unrelated protein in a monovalent fashion with ΔG_Non_ < ΔG_SP_, which can be enhanced if this non-specific binding occurs bivalently with 2× G_Non_. Namely, there will be a binding avidity of one antibody molecule to one protein. Such binding avidity is more pronounced when a binding assay is performed on surface-bound proteins, for example, including ELISA, dot blot, Western blot, and SPR, in which target proteins are attached to the surface (e.g., gel, membrane, polymer, and particle), as depicted in Figure 7. Unless the antigen is spaced in the correct orientation, an antibody would bind only one antigen with a free energy of ΔG_SP_. In an ideal situation, the antigen binding would become bivalent with the consequence of a two-fold binding energy 2× ΔG_SP_, which can lead to enhanced antigen binding. For non-specific binding, such a requirement of adequate spacing and correct orientation may not be essential and, hence, always with 2× ΔG_Non_, leading to non-specific binding [4]. It is also possible that an antibody binds to a surface-attached unrelated protein bivalently with 2× ΔG_Non_, although this mode of binding can occur for an unrelated protein, even in a solution. A more flexible antibody structure may lead to more bivalent binding to both target antigens and unrelated proteins.

## 6. Arginine

Arginine has been shown to play various roles in molecular interactions, including in intra-molecular packing, inter-molecular interactions, and lipid membrane interactions. These effects are mediated by electrostatic, π-cation, and aromatic/hydrophobic interactions [54,55,56,57]. Arginine is highly soluble in water, as is evident from its aqueous solubility (up to 2 M). Nevertheless, it has been demonstrated to have an affinity for aromatic compounds [55,58]. These aromatic materials are not only aromatic but also hydrophobic [59], indicating the possibility of arginine to have a hydrophobic property. It was shown that, while arginine overall increases the surface tension of water, the observed decline of the surface tension at high arginine concentration suggests the affinity of arginine for water [60]. These observations clearly demonstrate multiple interaction mechanisms that cannot be shared by other amino acid side chains. With regard to the antigen binding of CDR, it was shown that arginine contributes to non-specific binding or poly-reactivity. Unless it is positioned in CDR in the right context that can enhance binding (poly-reactivity against an antigen), it can generally cause non-specific binding [11]. The context-dependent effects of arginine have been shown to play a role in binding to macromolecules [11,12]. When placed next to Gly or Pro, arginine provided different types of molecular interactions. Gly offered a more flexible movement of arginine repeats and the random distribution of arginine side chains, while Pro offered a more extended structure and, hence, restricted the sdistribution of arginine side chains [61].

## 7. Conclusions

We have described two potential mechanistic insights into antibody poly-reactivity. The first mechanism revolves around the conformational changes that take place in antibody structures when exposed to an acidic environment. These changes encompass alterations in domain–domain interactions or even the unfolding of immunoglobulin domains, depending on the specific antibody. The flexibility of antibody structures is a key factor underlying these domain–domain interactions, and their visualization through individual structure measurements has provided valuable insights. Moreover, the bivalent nature of antibodies, coupled with their flexible domain–domain interactions, appears to play a crucial role in both augmented antigen binding and non-specific binding.

The second mechanism involves the introduction of arginine mutations into the complementarity-determining regions (CDRs) of antibodies. This strategic mutation not only enhances the binding mechanisms but also potentially leads to structural modifications in the antigen-binding regions. These findings shed light on the complexities of antibody poly-reactivity and contribute to a deeper understanding of their behavior in diverse physiological contexts. Further research in this direction will undoubtedly enhance our knowledge of antibody functioning and its implications for immunological responses.

## Figures and Tables

**Figure 1 antibodies-12-00064-f001:**
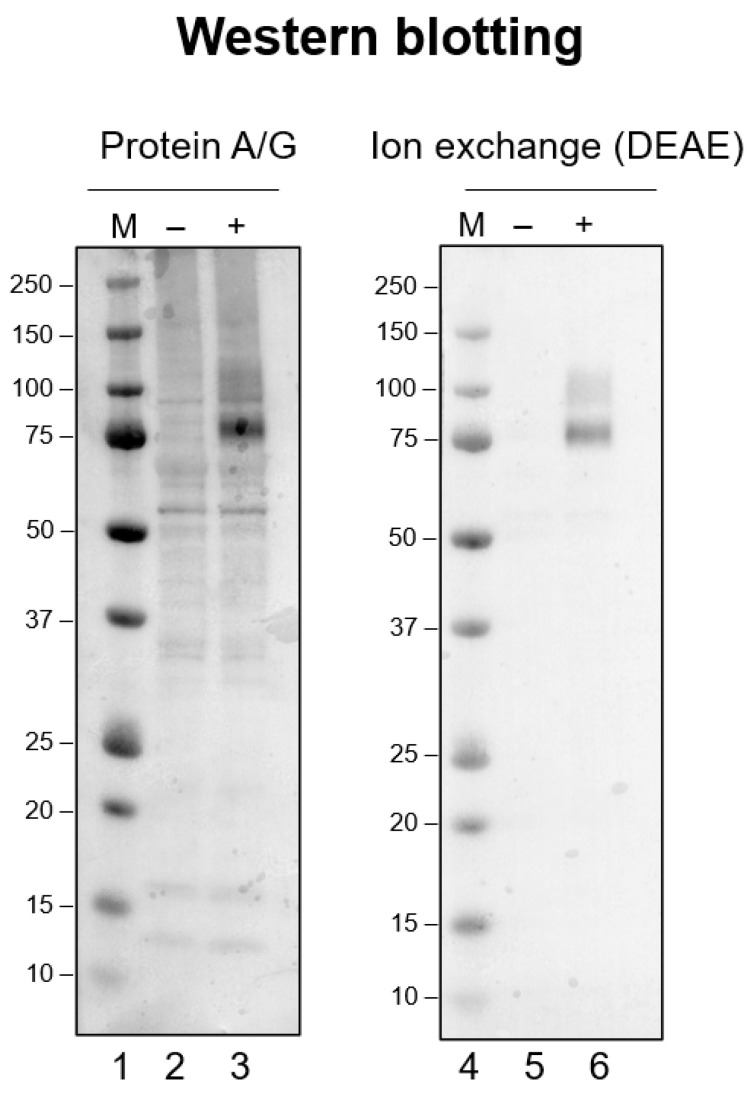
Comparison of Western blotting with anti-PLXDC2 rabbit monoclonal antibody prepared using two different purification methods. Immunoblotting analysis was performed using the whole cell lysates of HEK293 cells transfected with the human PLXDC2 (plexin-domain-containing protein 2) gene. The blots were probed with anti-PLXDC2 rabbit monoclonal antibody #4G3 (0.5 μg/mL) obtained through different purification methods. Cell lysates from untransfected and PLXDC2-transfected HEK293 cells are designated as “−” and “+”, respectively, while “M” represents the molecular weight markers. Left Panel: Western blot analysis using an antibody purified using protein A/G chromatography with acid elution. Right Panel: Western blot analysis using an antibody purified using CIMmultus DEAE chromatography at pH 8.0 throughout the chromatography. Both blots were exposed for an equal duration of 5 min.

**Figure 2 antibodies-12-00064-f002:**
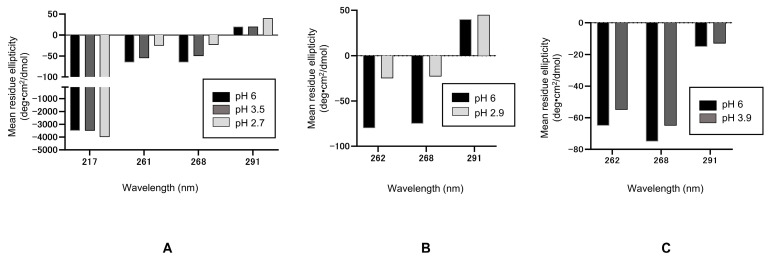
Summary of CD data for hIgG4−A (**A**), hIgG4−B (**B**), and murine antibody (**C**). These antibodies were exposed to low pH buffers.

**Figure 3 antibodies-12-00064-f003:**
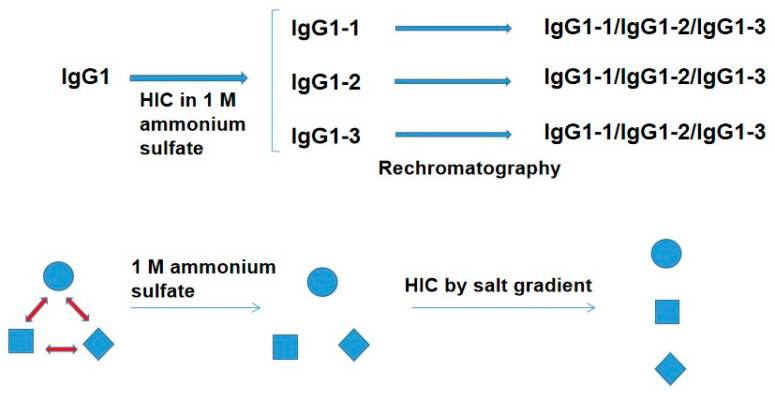
Schematic illustration of IgG1 behavior during HIC. IgG1 was made 1 M ammonium sulfate and bound to a HIC column. The bound IgG1 was eluted by descending ammonium sulfate gradient in the order of less hydrophobic (IgG1-1) to more hydrophobic (IgG1-3) antibody structures. Circle, diamond, and square represent different conformation in equilibrium in solution. These structures may be frozen by the addition of ammonium sulfate and separated using HIC.

**Figure 4 antibodies-12-00064-f004:**
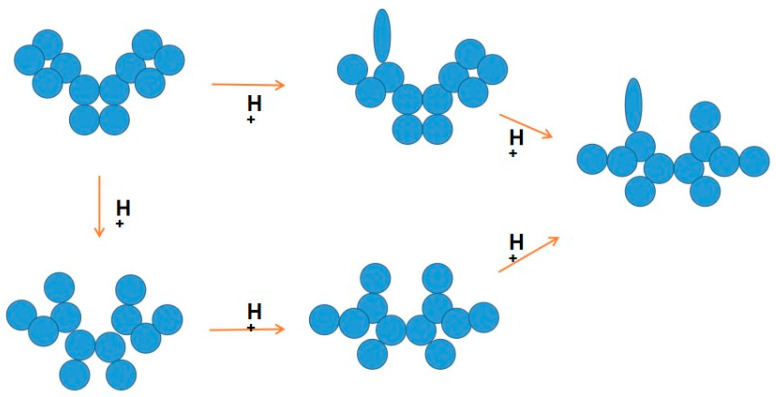
Potential structures of antibodies in acid. These structures represent different domain–domain interactions and at least one immunoglobulin domain unfolded.

**Figure 5 antibodies-12-00064-f005:**
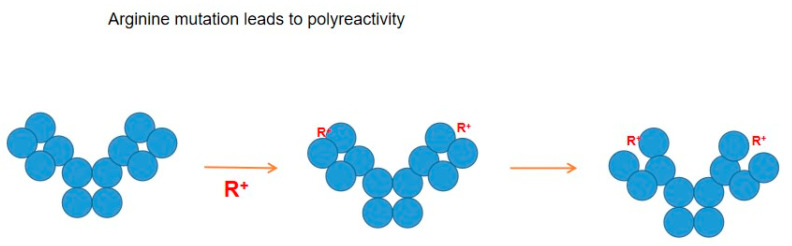
Potential structures caused by arginine mutation. Mutation to arginine (R^+^) not only adds a positive charge but also may alter domain–domain interactions.

**Figure 6 antibodies-12-00064-f006:**
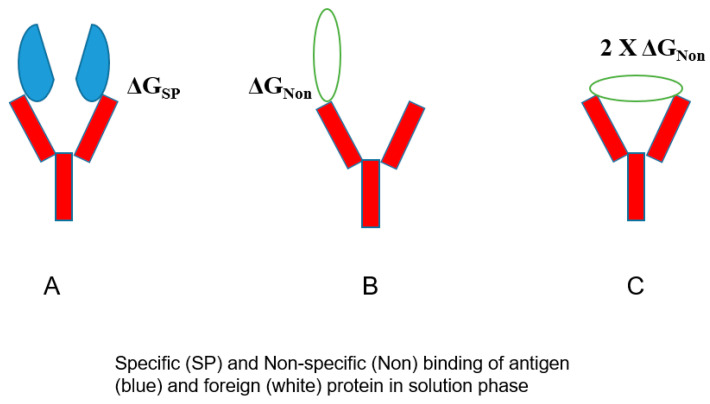
Antibody binding modes. (**A**) Bivalent binding of an antibody to two antigens. (**B**) Monovalent binding of an antibody to an unrelated protein. (**C**) Non-specific bivalent binding to an unrelated protein.

**Figure 7 antibodies-12-00064-f007:**
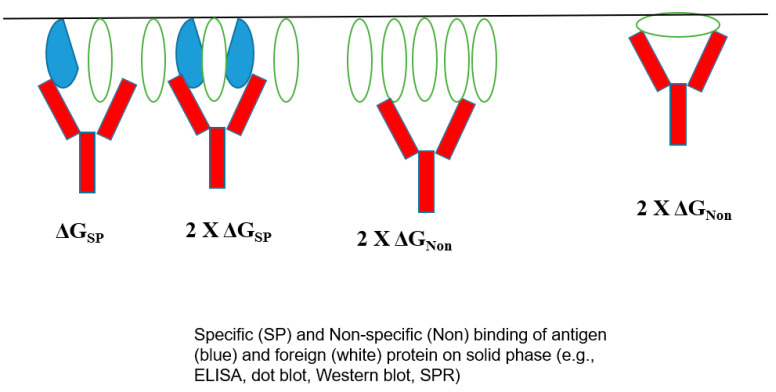
Potential modes of antibody binding to proteins attached to surface.

**Table 1 antibodies-12-00064-t001:** DSC analysis of hIgG4-A.

Antibodies	pH 6.0	pH 3.5	pH 2.7
P_I_ (minor peak)	67 °C	36 °C	UD
P_II_ (major peak)	78 °C	58 °C	41–49 °C

**Table 2 antibodies-12-00064-t002:** Sedimentation analysis of hIgG4-A.

pH	Main	Minor Peaks	Half-Antibody
pH 6.0	6.74 S (96.2%)	10–18.9 S (3.7%)	None
pH 3.5	7.08 S (97.5%)	10.9/13.1 S	3.5 S (1.6%)
pH 2.7	6.73 S (97.7%)	9.2 S < (2–3%)	3.6 S (0.9%)
pH 3.5 → pH 6.0	97.5%		
pH 2.7 → pH 6.0	67%	More oligomers	

## Data Availability

No new data were created or analyzed in this study. Data sharing is not applicable to this article.

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
