# Peer review of "Mechanistic Insight into Poly-Reactivity of Immune Antibodies upon Acid Denaturation or Arginine Mutation in Antigen-Binding Regions"

_2073-4468, 2023, doi:10.3390/antib12040064_

Round 1
Reviewer 1 Report
The manuscript “Mechanistic insight into poly-reactivity of antibodies upon acid denaturation or arginine mutation in antigen binding regions” represent comprehensive overview on the biochemical mechanism of pH induced polyreactivity of antibodies. The low pH can induce polyreactivity of human and mouse antibodies, which can influence their specificity and potentially their therapeutic activity.
The text is well written. Manuscript provides some experimental data which substantiate the review of the literature.
In addition to low pH polyreactivity of some antibodies can be induced by exposure to chaotropic agents and to high temperatures (see following publications: PMID: 11247642, PMID: 17097144, PMID: 18558357, PMID: 9692879, and PMID: 18558353). The manuscript will benefit if authors discuss these factors inducing polyreactivity and compare the effects with those observed after exposure to low pH.
Polyreactivity mediated by somatic mutagenesis is out of the scope of the manuscript that is mainly focused on low pH effects on antibodies Therefore this reviewer considers that it would be better this topic to be removed from the manuscript. Figure 5 should also be removed. Arg-mediated polyreactivity of antibodies can be subject of another article.
Figure 4 is not convincing and clear. There is not clear proof about protonation effect of low pH on antibodies as depicted. This figure and the associated discussion would be better to be removed from the text.
Author Response
We have added other conditions that lead to poly-reactivity. However, we decided to keep arginine mutation part, as that is one of the motivations to write this review.
Reviewer 2 Report
Τhe review by T. Arakawa and T. Akuta on “Mechanistic insight into poly-reactivity of antibodies upon acid denaturation or arginine mutation in antigen binding regions“ presents interesting aspects on mechanisms of antibody polyreactivity induction but it has nevertheless shortcomings.
The authors since addressing polyreactivity of antibodies should refer to specific publications of Prof. Stratis Avrameas, who was the first to isolate natural antibodies from the normal human serum and describe and demonstrate polyreactivity, at the monoclonal level. [DOI: 10.1016/j.jaut.2007.07.010, DOI: 10.1111/sji.12414]. Other researchers also delt with polyreactivity issues such as Abner Louis Notkins [DOI: 10.1093/infdis/jiu512, DOI: 10.1016/j.chom.2007.01.002, DOI:10.1016/j.it.2004.02.004]. Natural antibodies, i.e., germline or near germline encoded immunoglobulins found in all individuals without (known) prior antigenic experience, are by definition polyreactive; this is the marked difference from the classical “induced antibodies”, which are products of hyperimmunization with a specific antigen, expressing high affinity for it. In the context of the natural (endogenous) antibody repertoire, 'polyreactivity' is often defined as the ability of a mAb to bind a variety of self and foreign antigens which may be completely unrelated and is often attributed to 'a more conformationally flexible antigen binding pocket' recognizing 3D epitopes/charges (carrying an excess number of positive charges).
The authors should therefore strengthen their review by adding some additional introductory information: 1) clearly define the term “polyreactivity”, i.e., recognition of a variety of structurally unrelated molecules, such as proteins, nucleic acids, carbohydrates and haptens, 2) cite publications describing the particularities of the polyreactive binding site mainly CDR2 and CDR3 of natural monoclonal antibodies rich of basic aminoacids such as arginine, lysine, etc., and 3) emphasize the difference between antibodies, natural versus hyperimmune. The authors often mention the term “therapeutic” monoclonals most probably referring to hyperimmune ones, but this has to be clear to the readers.
Author Response
We have added poly-reactivity of general antibodies in serum.
Round 2
Reviewer 1 Report
Authors satisfactory addressed all my concern. The manuscript is appropriate for publication in its current form.
Author Response
No response is required.
Reviewer 2 Report
Τhere are still points that need to be clarified so that no ambiguity remains.
Specifically,
the ABSTRACT should be improved regarding its introductory sentences:
Poly-reactivity in the ABSTRACT has to be defined. [cross-reactivity/common epitopes of proteins primary sequence, or reactivity against structurally unrelated molecules (proteins, nucleic acids, haptens, etc), as NATURAL ANTIBODIES do?
ABSTRACT example
Poly-reactivity of an antibody is defined as…………… The poly-reactivity against structurally unrelated self and non-self antigens is an inherent property of naturally occurring antibodies present in normal serum, also demonstrated at the monoclonal level. Poly-reactivity can also be expressed by therapeutic antibodies, produced after hyperimmunization with an antigen, and plays a critical role in their clinical development. On the one hand, it can enhance their binding to target antigens and cognate receptors, …………….. (as it is)
Introduction/ line 33:
Naturally occurring antibodies have NOT “sticky properties”. This has been a very old objection, which has been definitively answered after a long period of experimentation and by many researchers worldwide. These antibodies bind specifically through their Fab portion, as also shown by competition ELISA with various antigens. Reactivity of low affinity, in vitro, does not mean “sticky” action. Regarding previous studies on polyreactivity, it has been shown that especially H-CDR3 of monoclonal polyreactive antibodies carrying an excess number of positive charges. This information should be discussed in this review in relation to their last section (6) on Arginine. (also discussed in references 8 and 9)
Title: Mechanistic insight into poly-reactivity of antibodies upon acid denaturation or arginine mutation in antigen binding regions
Comments: As the authors are interested in “therapeutic” and immune monoclonal antibodies, I would propose the addition of “immune /therapeutic” antibodies in the title, to differentiate them from naturally occurring ones.
In sections 2. and 3. of the review
For each monoclonal they present, the authors should always specify what the immunization antigen was; hyperimmune antibodies produced after hyperimmunization with a precise antigen.
Minor:
There are spelling errors in refs 8 and 9 (Avrameas instead of Abrameas; Ternynck instead of Terhynck
Τhere are other wording issues in English throughout the text.
Τhere are wording issues in English throughout the text.
Author Response
We have revised accordingly.

Round 3
Reviewer 2 Report
1. The authors should keep in the title only one of the two suggestions (immune or therapeutic) and in their case, I would keep immune antibodies.
2. In ABSTRACT, the sentence “The poly-reactivity can occur in natural serum antibodies likely due to their conformation flexibility and, for therapeutic antibodies, plays a critical role in their clinical development.” Does not make sense. Polyreactivity can be only demonstrated in natural monoclonal antibodies, as serum antibodies are polyclonal. Therefore, their reference to “therapeutic antibodies” within the same sentence is confusing as they probably mean the monoclonal immune antibodies widely used in therapy.
There are many spelling and grammatical errors throughout the manuscript, e.g.,
Line 313… Survaying literatures, we….
Line 455… upon pH netutralization…
Line 504…..arginine contribute…
Author Response
Thank you for valuable comments. We have addressed all the comments.
